# CogniLoad: A Synthetic Natural Language Reasoning Benchmark With Tunable Length, Intrinsic Difficulty, and Distractor Density

## Abstract

Current benchmarks for long-context reasoning in Large Language Models (LLMs) often blur critical factors like intrinsic task complexity, distractor interference, and task length. To enable more precise failure analysis, we introduce **CogniLoad**, a novel synthetic benchmark grounded in Cognitive Load Theory (CLT). CogniLoad generates natural-language logic puzzles with independently tunable parameters that reflect CLT's core dimensions: intrinsic difficulty ($d$) controls intrinsic load; distractor-to-signal ratio ($\rho$) manipulates extraneous load; and task length ($N$) serves as an operational proxy for conditions demanding germane load. Evaluating 14 SotA reasoning LLMs, CogniLoad reveals distinct performance sensitivities, identifying task length as a dominant constraint and uncovering varied tolerances to intrinsic complexity and U-shaped responses to distractor ratios. By offering systematic, factorial control over these cognitive load dimensions, CogniLoad provides a reproducible, scalable, and diagnostically rich tool for dissecting LLM reasoning limitations and guiding future model development.

## 1 Introduction

Cognitive Load Theory (CLT) [Sweller, 1988] posits that working memory constraints [Lieder and Griffiths, 2020] for problem solving in humans arise from three types [Paas et al., 2003] of cognitive load: intrinsic (ICL), extraneous (ECL), and germane (GCL). ICL stems from the inherent complexity and element interactivity of the task [Halford et al., 1998]. ECL is induced by suboptimal task presentation requiring the processing of elements that are not task-relevant [Chandler and Sweller, 1991]. GCL pertains to remaining resources effectively allocated to engaging with the intrinsic task demands for schema construction [Ericsson and Kintsch, 1995, Sweller, 2010].

Large language models (LLMs) face analogous demands on their finite computational resources. The essential element interactivity of a reasoning chain mirrors ICL; distractor elements reflect ECL; and sustained engagement with intrinsically relevant information over a long reasoning process acts as an operational proxy for germane-like processing - the constructive effort to maintain a coherent problem representation.

To the best of our knowledge, no study has based the evaluation of problem-solving capacities of LLMs in CLT by distinguishing these three load types, and existing benchmarks often confound them: LongBench [Bai et al., 2024a] and L-Eval [An et al., 2024] vary context length but not necessarily the intrinsic reasoning depth; LogicBench [Parmar et al., 2024] probes ICL with minimal demands on ECL or context-induced load; BABILong [Kuratov et al., 2024] mixes multi-step reasoning with fixed distractor ratios, obscuring precise failure attribution.

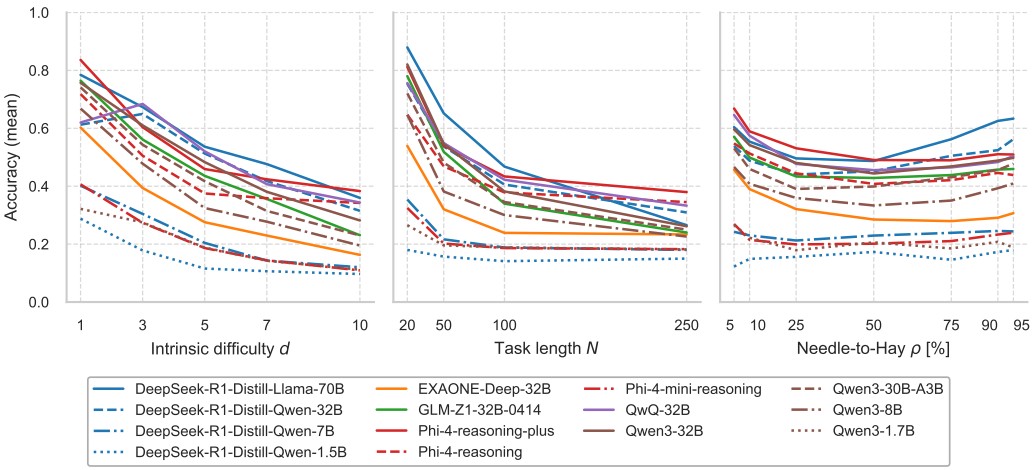

Figure 1: The average accuracy of models across the evaluated parameter space for $d \in \{1, 3, 5, 7, 10\}$ (left panel), $N \in \{20, 50, 100, 250\}$ (center panel), and $\rho \in \{5, ..., 95\}$ (right panel). Each plot selects one dimension for the X-axis and averages the accuracy of all evaluated puzzles for the other two dimensions relative to it.

We introduce **CogniLoad**, a controllable synthetic benchmark for long-context reasoning, guided by CLT, that operationalizes these load types through tunable parameters in randomized natural-language logic puzzles: **(i) Intrinsic Load** via Intrinsic Difficulty $d$ controls the number of interacting entities, attributes, and logical clauses, directly manipulating ICL by varying essential element interactivity and reasoning depth. **(ii) Extraneous Load** via Distractor Density $\rho$: Dictates distractor density; lower $\rho$ increases irrelevant elements, manipulating ECL. **(iii) Germane Load Proxy** via Task Length $N$ serves as an operational proxy for demanding germane-like cognitive work.

In this study we make the following contributions:

1. We ground the evaluation of LLMs in CLT, precisely defining benchmark parameters that control ICL, ECL, and an operational proxy for the conditions conducive to GCL.

2. We introduce *CogniLoad*, the first benchmark designed to independently control these three dimensions of cognitive load, while scaling to arbitrarily long contexts.

3. We provide an algorithm for the automatic randomized generation and evaluation of puzzle instances, enabling large-scale and reproducible comparison of LLM capabilities.

4. We report empirical results on 14 state-of-the-art (SotA) reasoning LLMs (see Figure 1), revealing distinct failure regimes across the $(d, N, \rho)$ dimensions and highlighting specific targets for improving LLM design.

Together, these contributions translate CLT into a precise diagnostic framework for understanding and advancing long-context reasoning in LLMs.

## 1.1 Related work

**Long-Context Benchmarks (Working Memory Capacity).** A line of work starting with Long-Range Arena (LRA) [Tay et al., 2020] and followed by several recent benchmarks probe LLM performance on long sequences, often framed as testing "memory load" or context utilization. Earlier studies such as SCROLLS [Shaham et al., 2022], BookSum [Kryściński et al., 2021], and QMSum [Zhong et al., 2021] scale document length without manipulating intrinsic difficulty. LongBench [Bai et al., 2024a,b] and L-Eval [An et al., 2024] aggregate multi-task corpora up to 200k tokens, while BABILong [Kuratov et al., 2024], LongReason [Ling et al., 2025], RULER [Hsieh et al., 2024], ZeroSCROLLS [Shaham et al., 2023], and Michelangelo [Vodrahalli et al., 2024] increase context but the inherent difficulty of individual sub-tasks (ICL) may vary unsystematically while distractor density (ECL) is often not a controlled variable. Consequently, performance degradation could be due

to sheer length overwhelming processing capacity, or an inability to sustain germane-like cognitive work over extended relevant information, but the precise cause of failure is not clear.

**Logical-Reasoning Benchmarks (Intrinsic Load).** A complementary line of benchmarks focuses on ICL by presenting tasks with high inherent complexity but often within minimal context lengths or distractors. Notable classical suites include ReClor [Yu et al., 2020], LogiQA [Liu et al., 2020], and BIG-Bench-Hard (BBH) [Suzgun et al., 2022]. AutoLogic [Zhu et al., 2025] is a benchmark that explicitly focuses on scaling ICL through controllable complexity. LogicBench [Parmar et al., 2024], CLUTRR [Sinha et al., 2019], and ZebraLogic [Lin et al., 2025] also exemplify this by formulating symbolic logic puzzles that demand processing many interacting elements (e.g., multi-step deductions, handling negation, constraint satisfaction). Similarly, mathematical reasoning datasets like GSM8K [Cobbe et al., 2021] and abstract rule induction tasks like ARC-AGI [Chollet et al., 2024] primarily escalate ICL by increasing the complexity of essential rules and their interdependencies.

**Needle-in-A-Haystack Benchmarks (Extraneous Load).** Needle-in-a-Haystack (NIAH) designs [Gkamradt, 2023] specifically target ECL by embedding relevant facts ("needles") within large volumes of distractor text ("hay"). Variants like Sequential NIAH [Yu et al., 2025] and Nolima [Modarressi et al., 2025] investigate the impact of such distractors, which constitute non-essential elements requiring processing for filtering, thereby imposing ECL. While these benchmarks effectively isolate the impact of distractors on information retrieval, the "needle" tasks themselves typically involve low ICL (e.g., simple fact lookup).

**Need for Multi-Dimensional Evaluation.** CLT highlights the interplay of ICL, ECL, and germane processing within finite working memory [Paas et al., 2003]. Existing LLM reasoning benchmarks, however, typically manipulate only one dimension without systematic, independent control over the others. Even benchmarks like MIR-Bench [Yan et al., 2025], which combine high ICL with extensive input, do not offer the factorial control needed to disentangle these loads, hindering precise diagnostics.

**Contribution of CogniLoad.** CogniLoad addresses this critical gap by providing a framework for independently controlling parameters that influence: (i) ICL via intrinsic puzzle difficulty ($d$), (ii) ECL via distractor density ($\rho$), and (iii) the demands for sustained, germane-like processing via task length ($N$), all within a single synthetically generated natural language puzzle. This factorial design enables a precise diagnosis of LLM failure modes — for instance, determining whether performance degradation at long contexts stems from an inability to handle increased intrinsic complexity, susceptibility to extraneous distractors, or an incapacity to maintain coherent reasoning over extended sequences. By explicitly grounding these dimensions in Cognitive Load Theory, CogniLoad offers the first benchmark to diagnostically map LLM capability surfaces across these distinct cognitive demands, thereby complementing and extending the insights from evaluations that focus on single factors.

## 2 Benchmark Design: CogniLoad Logic Puzzles

CogniLoad is a family of natural-language logic-grid puzzles expressly crafted to probe sequential reasoning capabilities of LLMs. The design goals are threefold: each puzzle (i) necessitates sequential multi-step deduction where order fundamentally matters; (ii) embeds a controllable number of relevant "needle" facts within the context of a controllable number of "hay" distractor statements; and (iii) provides parameters that control distinct dimensions of cognitive load. This section formalizes the task, describes the puzzle generation process, details the control parameters, and motivates key design choices.

### 2.1 Puzzle Definition

Each puzzle in CogniLoad (see Figure 2 for an example) consists of a set of people with independent and mutable attributes. A series of statements, applied in strictly sequential order, updates these attributes according to conditions specified in each statement. The puzzle generation is parameterized by the three key parameters: the intrinsic difficulty $d$, the total number of statements $N$, and the needle-to-hay ratio $\rho$.

> **(i) Puzzle Instruction:** Solve this logic puzzle. You MUST finalize your response with a single sentence about the asked property (e.g., "Peter is in the livingroom.", "Peter is wearing blue socks",... ). Solve the puzzle by reasoning through the statements in a strictly sequential order.
>
> **(ii) Initial State:**
> - Brent is wearing green socks and is wearing purple gloves and last listened to classical music.
> - Anthony is wearing purple socks and is wearing yellow gloves and last listened to disco music.
> - ...
>
> **(iii) Update Statements:**
> 1. The people wearing green socks listen to electronic music.
> 2. The people who last listened to classical music and wearing purple gloves put on yellow gloves.
> 3. ...
>
> **(iv) Query:** What color of socks is Brent wearing?

Figure 2: Example CogniLoad puzzle with intrinsic difficulty $d = 3$, statements $N = 20$, and needle-to-hay ratio $\rho = 50\%$. Only a subset of the initial state and update statements is shown.

### 2.1.1 Basic Elements

A puzzle is formally characterized by the following components:

- **People**: A set $P = \{p_1, p_2, \ldots, p_n\}$ of persons in the puzzle, and $n = \max(d, 2)$.
- **Person of Interest (PoI)**: A randomly selected person $p^* \in P$ about whom the final question is asked.
- **Attribute Categories**: A set $A = \{c_1, c_2, \ldots, c_d\}$ of attributes randomly selected from a predefined taxonomy of 12 categories. Each category takes values in a Value Domain with a given finite cardinality, larger or equal to 10.
- **Value Domains**: For each category $c \in A$, a value domain $V_c = \{v_{c,1}, v_{c,2}, \ldots, v_{c,\ell_c}\}$ where $\ell_c = d + 1$ for $d > 1$ or $\ell_c = 3$ when $d = 1$. See Table 1 for examples.
- **State Function**: $S_t(p, c)$ representing the value of attribute $c$ for person $p$ at step $t$. Each person has values for the $d$ attribute of the selected attribute categories $A$, thus the state value represents a vector of dimension $d$.

Table 1: Overview of the attribute ontology. The full ontology contains 12 categories of varying domain sizes and is detailed completely in the Supplementary Material.

| Category Name (Code) | Domain Size | Examples of Values |
|---|---|---|
| location | 50+ | kitchen, balcony, zoo, museum, park... |
| clothes_socks | 10 | blue, red, yellow, green, purple... |
| clothes_gloves | 10 | (same as clothes_socks) |
| hair | 10 | (same as clothes_socks) |
| recent_listen | 13 | rock, jazz, disco, classical, funk... |
| recent_eat | 10 | pizza, pasta, burrito, sushi, taco... |
| ... | ... | ... |

### 2.1.2 Initialization

The puzzle starts with initialization statements ($t = 0$) that assigning unique attribute values to each person: $\forall p \in P, \forall c \in A : S_0(p, c) \in V_c$ such that $\forall p_i, p_j \in P, i \neq j, \exists c \in A : S_0(p_i, c) \neq S_0(p_j, c)$.

### 2.1.3 Statement Generation Process

For each step $t$ from 1 to $N$, a statement is generated that changes the state of a person. If it updates the PoI, the statement is called a *needle* and for a non-PoI it is called a *hay*.

1. **Statement Type Selection**: Given $N$ and $\rho$, let $n_{\text{needle}}^t$ and $n_{\text{hay}}^t$ be the remaining numbers of needles and hays to generate, to guarantee the desired proportion $\rho$ in the complete puzzle. The probability of

selecting a needle statement is then $\mathbb{P}(T_t = \text{needle}) = n_{\text{needle}}^t/(N - t)$. The total number of needle statements in the puzzle is calculated as $n_{\text{needle}}^0 = \max(1, \min(N, \text{round}(N \cdot \rho/100)))$.

2. **Reference Person Selection**: Given the selected statement type $T_t$, the algorithm selects the reference person $r_t$: if $T_t = \text{needle} \implies r_t = p^*$ and if $T_t = \text{hay} \implies r_t \sim \text{Uniform}(P \setminus \{p^*\})$.

3. **Statement Structure**: For each statement sample a number of conditions $k_t \sim \text{Uniform}\{1, \ldots, d\}$, and a number of state updates $m_t \sim \text{Uniform}\{1, \ldots, d\}$ and uniformly sample attribute categories $C_t \subseteq A, \; |C_t| = k_t$ and state updates $U_t \subseteq A, \; |U_t| = m_t$.

4. **Condition and Update Value Specification**: For each category $c \in C_t$, the condition value is determined by the reference person's current state: $v_{c,t} = S_{t-1}(r_t, c)$. For needles these conditions target the PoI, for hays the conditions can match multiple people. For update values if $T_t = \text{needle} \implies u_{c,t} \sim \text{Uniform}(V_c)$ and if $T_t = \text{hay} \implies u_{c,t} \sim \text{Uniform}(V_c \setminus \{S_{t-1}(p^*, c)\})$.

5. **Logical Form**: The statement at step $t$ has the logical form:

$$\forall p \in P : \left( \bigwedge_{c \in C_t} S_{t-1}(p, c) = v_{c,t} \right) \Rightarrow \left( \bigwedge_{c \in U_t} S_t(p, c) = u_{c,t} \right).$$

Attributes not mentioned in the update set remain unchanged $\forall p \in P, \forall c \in A \setminus U_t : S_t(p, c) = S_{t-1}(p, c)$. This is not specified in the prompt but implicitly assumed by the LLMs.

### 2.1.4 Validation Constraints

A sequence of validations verifies that the generated statement does not result in a state that prevents the generation of further needles and hays. If all validations pass, the statement is appended to the puzzle; otherwise a new statement is generated.

*For hay statements ($r_t \neq p^*$):* After the update, the state of affected non-PoIs must not become identical to PoI $\forall p \in P \setminus \{p^*\}$ such that $\forall c \in C_t : S_{t-1}(p, c) = v_{c,t}, \exists c \in A : S_t(p, c) \neq S_t(p^*, c)$ and the update must not affect the PoI $\exists c \in C_t : S_{t-1}(p^*, c) \neq v_{c,t}$.

*For needle statements ($r_t = p^*$):* The update must not affect all non-PoI people $\exists p \in P \setminus \{p^*\} : \exists c \in C_t : S_{t-1}(p, c) \neq v_{c,t}$ and after the update not all non-PoIs can equal the PoI $\exists p \in P \setminus \{p^*\} : \exists c \in A : S_t(p, c) \neq S_t(p^*, c)$.

To prevent the distractors from becoming too trivial to track at lower difficulties we further validate that a hay statement does not result in all non-PoIs becoming identical so the set $P \setminus \{p^*\}$ must contain at least two persons with distinct attribute values. As a consequence of the algorithm design, the hay statement $T_t = \text{hay}$ by definition must affect at least one non-PoI $\exists p \in P \setminus \{p^*\} : \forall c \in C_t : S_{t-1}(p, c) = v_{c,t}$.

### 2.1.5 Final Question Generation

After all $N$ statements have been generated, the puzzle concludes with a question about a random attribute of the PoI, sampled as a random category $c_q \sim \text{Uniform}(A)$. The correct answer to the puzzle is $S_N(p^*, c_q)$ obtained from the final state of the PoI.

### 2.1.6 Evaluation metrics

We evaluate the success of the solver $M$ based on the exact string match of the final queried attribute value in the last two sentences of the response. For each puzzle instance $z \in Z$ from our evaluation set $Z$, we compare the model's answer ($\text{answer}_M(z)$) with the true value of the attribute derived from the final state of the PoI. The accuracy of a model $M$ across the evaluation set is calculated as $\text{acc}(M) = \frac{1}{|Z|} \sum_{z \in Z} \mathbf{1} \left[ \text{answer}_M(z) = S_N(p^*, c_q) \right]$ where $S_N(p^*, c_q)$ represents the final state value of the queried attribute $c_q$ for the PoI $p^*$ after all $N$ statements have been processed. This value is computed by our puzzle generation algorithm.

## 2.2 Tunable Parameters

To systematically probe different facets of long-context reasoning, the CogniLoad generator employs three independent parameters. These parameters are designed to operationalize distinct cognitive

Table 2: Key parameters controlling the puzzle generation.

| Symbol | Name | Definition | Cognitive Load Affected |
|--------|------|------------|-------------------------|
| $d$ | Intrinsic Difficulty | Controls cardinality of people set $|P| = \max(d, 2)$, attribute categories $|A| = d$, for each category $c \in A$ the cardinality of value domains $|V_c| = \max(d+1, 3)$, and the distribution of conditions and updates per statement: $k, m \sim \text{Uniform}\{1, ..., d\}$. | ICL: Element interactivity, state space/rule complexity. |
| $N$ | Task Length | Total number of sequential state transitions in the puzzle. | GCL Proxy / Task Length: Demands sustained engagement with core elements. |
| $\rho$ | Needle-to-Hay Ratio | Percentage of statements directly influencing the PoI (needles) versus distractor statements (hay) | ECL: Distractor density challenges filtering, selective attention, and imposing load from processing non-essential elements. |

load dimensions as defined by CLT [Paas et al., 2003], allowing the creation of puzzles with varying characteristics. Together, they define the load profile of a puzzle instance.

**Intrinsic Difficulty** ($d$)   for $d \in \{1, 3, 5, 7, 10\}$ controls multiple facets of puzzle complexity (see Table 2), directly manipulating ICL which according to CLT hinges on element interactivity [Halford et al., 1998]. Higher $d$ increases ICL via: (i) combinatorial growth in state space ($\approx (d+1)^d$), (ii) increased interactivity between persons, attributes, and values, and (iii) increased rule complexity (up to $d$ conditions/updates per statement).

**Task Length** ($N$)   for $N \in \{20, 50, 100, 250\}$ sets the total number of sequential state-update statements. While directly determining sequence length, $N$ serves as an operational proxy for conditions demanding GCL. Higher $N$, particularly with high $d$ (intrinsic difficulty) and high $\rho$ (relevance), compels deeper reasoning through more essential interacting elements [Sweller, 2010]. Additionally, higher $N$ also necessitates the maintenance of a coherent (stateful) problem representation over a longer term with the construction of an efficient schema for it [Ericsson and Kintsch, 1995].

**Needle-to-Hay Ratio** ($\rho$)   for $\rho \in \{5, ..., 95\}$ sets the percentage of PoI-relevant ("needle") versus distractor ("hay") statements, directly manipulating ECL. ECL arises from processing non-essential elements [Chandler and Sweller, 1991]. Lower $\rho$ increases ECL via higher distractor density which challenges filtering. Higher $\rho$ reduces ECL by focusing resources on relevant information. Critically, CogniLoad's "hay" statements are syntactically similar to "needles" and involve valid state updates for non-PoIs, imposing a more challenging ECL than easy to distinguish distractor text.

## 3   Results

We evaluated the performance of 14 LLMs on 100 random CogniLoad puzzles per $(d, N, \rho)$ configuration resulting in 14'000 puzzle instances per LLM in total. We attempted to include every currently available Open-Weights LLM that is specifically trained for reasoning, but the VRAM limitations of our single-node inference environment (i.e. AMD MI250X accelerators) prevents us from evaluating the full DeepSeek-R1 model with 685B parameters.

Figure 1 shows mean accuracy across models as each load dimension varies with trends corroborated by our regression analysis (Section 3.1).

**Intrinsic difficulty** ($d$) Performance generally declines monotonically with $d$. For instance, even top models show a significant drop between $d = 1$ and $d = 3$, while degradation is less pronounced

Table 3: Per-model quadratic-$\rho$ GLM estimates with Wald $z$ statistic for p-values alongside derived 50% load-capacity thresholds (see Section 3.1.3). The value $--$ for $\mathrm{NT}_{50}$ indicates that no real root exists in $[0, 1]$. "DS" abbreviates "DeepSeek-R1-Distill" in the model names. $^{***}p<0.001$, $^{**}p<0.01$, $^{*}p<0.05$

| Model | $\beta_0$ | $\beta_d$ | $\beta_N$ | $\beta_\rho$ | $\beta_{\rho^2}$ | $\mathrm{ECL}_{50}$ | $\mathrm{NT}_{50}$ | $\mathrm{ID}_{50}$ |
|---|---|---|---|---|---|---|---|---|
| DS-Llama-70B | $7.83^{***}$ | $-0.27^{***}$ | $-3.10^{***}$ | $-3.41^{***}$ | $3.80^{***}$ | 66.9 | 0.6 | 4.92 |
| DS-Qwen-32B | $4.58^{***}$ | $-0.18^{***}$ | $-1.88^{***}$ | $-2.01^{***}$ | $2.24^{***}$ | 51.1 | 0.8 | 3.71 |
| DS-Qwen-7B | $1.47^{***}$ | $-0.20^{***}$ | $-0.93^{***}$ | $-0.46$ | $0.55$ | 2.3 | $--$ | $-1.71$ |
| DS-Qwen-1.5B | $-0.72^{***}$ | $-0.16^{***}$ | $-0.23^{***}$ | $0.72^{*}$ | $-0.41$ | 0.0 | $--$ | $-5.41$ |
| Phi-4-reasoning-plus | $5.61^{***}$ | $-0.23^{***}$ | $-1.91^{***}$ | $-3.10^{***}$ | $2.40^{***}$ | 63.7 | 0.88 | 4.83 |
| Phi-4-reasoning | $3.50^{***}$ | $-0.18^{***}$ | $-1.24^{***}$ | $-2.40^{***}$ | $1.95^{***}$ | 32.4 | 0.14 | 2.81 |
| Phi-4-mini-reasoning | $1.36^{***}$ | $-0.21^{***}$ | $-0.77^{***}$ | $-1.71^{***}$ | $1.68^{***}$ | 0.6 | $--$ | $-2.47$ |
| QwQ-32B | $5.00^{***}$ | $-0.18^{***}$ | $-1.80^{***}$ | $-3.52^{***}$ | $2.92^{***}$ | 49.2 | 0.93 | 3.6 |
| EXAONE-Deep-32B | $3.90^{***}$ | $-0.25^{***}$ | $-1.49^{***}$ | $-3.31^{***}$ | $2.57^{***}$ | 11.6 | $--$ | 0.55 |
| GLM-Z1-32B-0414 | $7.05^{***}$ | $-0.32^{***}$ | $-2.72^{***}$ | $-3.01^{***}$ | $2.56^{***}$ | 46.5 | 0.14 | 3.65 |
| Qwen3-32B | $7.12^{***}$ | $-0.29^{***}$ | $-2.72^{***}$ | $-3.21^{***}$ | $2.74^{***}$ | 53.7 | 0.94 | 4.08 |
| Qwen3-30B-A3B | $5.80^{***}$ | $-0.29^{***}$ | $-2.23^{***}$ | $-3.01^{***}$ | $2.91^{***}$ | 36.8 | 0.99 | 3.03 |
| Qwen3-8B | $4.88^{***}$ | $-0.27^{***}$ | $-1.95^{***}$ | $-2.99^{***}$ | $2.77^{***}$ | 23.7 | $--$ | 1.76 |
| Qwen3-1.7B | $0.57^{***}$ | $-0.16^{***}$ | $-0.46^{***}$ | $-1.48^{***}$ | $1.19^{***}$ | 0.0 | $--$ | $-4.5$ |

beyond $d = 7$, suggesting diminishing marginal effects of this complexity type for many models. At $d = 5$, 10 of 14 models are wrong in more than 50% of the puzzles.

**Memory load** ($N$) Memory load exhibits the steepest performance decline, with a substantial drop observed for most models between $N = 20$ and $N = 50$. This underscores the role of task length as a proxy for germane load as a primary contributor to cognitive load.

**Extraneous load** ($\rho$) Extraneous load often exhibits a U-shaped response, with performance minima typically around $\rho = 25 - 50\%$. However, the curve's depth and recovery at high $\rho$ vary significantly between models. Interestingly, DS-Llama-70B fully recovers and exceeds its initial performance ($0.60 \rightarrow 0.63$) while Phi-4-reasoning-plus shows only a partial recovery ($0.67 \rightarrow 0.51$).

## 3.1 Load–sensitivity Regression

To quantify model-specific sensitivities of the accuracy to load dimensions and derive interpretable capacity thresholds for each model, we employ a regression-based approach that allows us to isolate the impact of each type of cognitive load (see Table 3).

### 3.1.1 Regression Model Specification

We model the performance of LLMs using a binomial generalized linear model (GLM) with a logit link function:

$$\Pr(Y{=}1) = \sigma\big(\beta_0 + \beta_d\, d + \beta_N\, \log_{10} N + \beta_\rho\, \rho + \beta_{\rho^2}\, \rho^2\big),$$

where the binary outcome $Y$ represents exact-match accuracy ($Y = 1$, when the model solves the puzzle correctly), $\sigma(\cdot)$ is the inverse logit function, and the coefficients $\beta_d$, $\beta_N$ and $\beta_\rho$ quantify sensitivity to intrinsic difficulty (ICL), task length (GCL), and distractor ratios (ECL), respectively. The inclusion of a quadratic term for $\rho$, with the coefficient $\beta_{\rho^2}$, is motivated by the characteristic U-shape observed in the third panel of Figure 1 and based on an improved Akaike Information Criterion (AIC) value for 14 out of the 15 fitted models when included (see Supplementary Material). Since $N$ ranges up to 250, we apply $\log_{10}$ to keep it at a similar scale as the other parameters of the regression.

### 3.1.2 Significance of Main Effects

In all models, $\beta_d$ and $\beta_N$ are significant and highly negative, confirming performance degradation with increased intrinsic cognitive load and task length. The quadratic term for $\rho$ is also significant (except for two models) confirming the U-shaped response for most models: models typically perform worst at intermediate relevance ratios and recover as $\rho$ approaches either extreme. Two models (DS-Qwen-1.5B, DS-Qwen-7B) exhibit exhibit statistically insignificant coefficients for $\rho$

terms, likely reflecting their poor baseline performance rather than a genuine lack of association with the needle/hay ratio.

### 3.1.3 Capacity Points at 50% Accuracy

The GLM coefficients (Table 3) allow us to derive interpretable capacity thresholds. These represent the point at which a model's accuracy is predicted to drop to 50% when varying a single load parameter, while holding other load parameters at their estimated mean values:

**$\text{ECL}_{50}$** (Effective Context Length): Maximum number of statements a model can process while maintaining 50% accuracy. Higher values indicate superior context handling.

**$\text{NT}_{50}$** (Needle-to-hay Threshold): Minimum proportion of relevant information required to maintain 50% accuracy. Crucially, *lower* values indicate greater robustness to distractors. If the estimated $\text{NT}_{50}$ is missing, then the model accuracy is not expected to cross the 50% threshold for any value $0 \leq \rho \leq 1$, under mean conditions for $d$ and $N$.

**$\text{ID}_{50}$** (Intrinsic Difficulty): It is the maximum intrinsic complexity (number of interacting entities/attributes) that a model can handle while maintaining 50% accuracy. Negative values indicate failure to reach 50% accuracy even at the lowest difficulty setting under mean conditions for $N$ and $\rho$.

Mathematically, these thresholds are derived by setting the logit in the GLM equation to zero (for $\Pr(y = 1) = 0.5$) and solving for the parameter of interest, e.g.:

$$\text{ECL}_{50} = 10^{-(\beta_0 + \beta_d \bar{d} + \beta_\rho \bar{\rho} + \beta_{\rho^2} \bar{\rho}^2)/\beta_N}; \quad \text{ID}_{50} = -(\beta_0 + \beta_N \overline{\log_{10} N} + \beta_\rho \bar{\rho} + \beta_{\rho^2} \bar{\rho}^2)/\beta_d.$$

For $\text{NT}_{50}$, we solve the quadratic equation $\beta_0 + \beta_d \bar{d} + \beta_N \overline{\log_{10} N} + \beta_\rho \rho + \beta_{\rho^2} \rho^2 = 0$ for $\rho$.

### 3.1.4 Model Capacity

The regression analysis and estimated capacity thresholds (Table 3) reveal clear variations among models that can be grouped into three classes:

*High-Capacity Models:* DS-Llama-70B ($\text{ECL}_{50}$=66.9, $\text{ID}_{50}$=4.92) and Phi-4-reasoning-plus ($\text{ECL}_{50}$=63.7, $\text{ID}_{50}$=4.83) demonstrate exceptional context length tolerance and robust reasoning capabilities across all dimensions.

*Mid-Capacity Models:* Models such as DS-Qwen-32B ($\text{ECL}_{50}$=51.1), Qwen3-32B ($\text{ECL}_{50}$=53.7), QwQ-32B ($\text{ECL}_{50}$=49.2), and GLM-Z1-32B-0414 ($\text{ECL}_{50}$=46.5) constitute a middle tier. Their $\text{ID}_{50}$ values typically fall between 3.5 and 4.1, suggesting competence on problems of moderate complexity and length.

*Low-Capacity Models:* Smaller models, particularly DS-Qwen-1.5B and Qwen3-1.7B, exhibit minimal effective context handling capacity ($\text{ECL}_{50}$=0.0) and negative $\text{ID}_{50}$ values. This indicates that they fail to achieve 50% accuracy even at baseline difficulty and mean context/distractor levels, deteriorating rapidly under any increasing load.

### 3.1.5 Differential Sensitivity to Load Dimensions

The estimated coefficients further reveal distinct sensitivity profiles:

*Sensitivity to context length* ($\beta_N$): Universally negative and potent, with larger models often showing greater relative degradation from their higher baselines.

*Sensitivity to intrinsic difficulty* ($\beta_d$): Negative across models, but with a narrow range suggesting a more uniform effect.

*Sensitivity to information relevance* ($\beta_\rho$ and $\beta_{\rho^2}$): Confirms the U-shaped response, but $\text{NT}_{50}$ values reveal nuanced distractor robustness differences masked by aggregate scores (e.g., DS-Llama-70B vs. Qwen3-32B).

## 4 Discussion

CogniLoad, by operationalizing CLT, enables a multi-dimensional evaluation of LLM reasoning, revealing nuanced failure patterns obscured by single-dimension benchmarks. Our empirical results

(Section 3) offer several key insights: task length ($N$) emerges as a dominant determinant, suggesting challenges in sustained, germane-like processing for long, intrinsically demanding tasks; models exhibit distinct sensitivities to intrinsic difficulty ($d$) versus extraneous load ($\rho$), with the latter surprisingly showing U-shaped performance curves, indicating particular difficulties with intermediate distractor densities, while performing better for lowest and highest needle-to-hay proportions; and estimated capacity thresholds provide concise "cognitive fingerprints" for diagnostic LLM evaluation.

The limitations of our study are important to emphasize:

**Nuances of the CLT-LLM Analogy** While CLT provides a powerful analogous framework, it is crucial to acknowledge that "cognitive load" in LLMs manifests as computational constraints (e.g., attention saturation, representational bottlenecks) rather than biological working memory limitations. Our operationalization of $N$ as a proxy for conditions demanding GCL, for example, is an abstraction. Future research should aim to bridge CLT concepts with direct, mechanistic measures of LLM computational processes to refine this analogy and deepen our understanding of artificial cognition.

**Scope of Reasoning and Generalizability** CogniLoad currently focuses on sequential, deductive logic-grid puzzles. This controlled environment enables precise manipulation of load factors, but the extent to which these specific load sensitivities generalize to other reasoning paradigms (e.g., abductive, inductive, mathematical, commonsense) remains an open question. Extending the CLT-grounded multi-dimensional evaluation to diverse reasoning domains is a promising next step.

**Beyond Accuracy and Main Effects** The current evaluation relies on simple exact-match accuracy. Future iterations could incorporate richer metrics (e.g., step-wise reasoning fidelity, solution coherence, uncertainty of solutions) and systematically investigate interaction effects between $d, N, \rho$, which CogniLoad's factorial design supports.

**Architectural Implications** Pinpointing the specific decisions in LLM architecture and training regimes that result in our observed performance differential requires thorough analysis and experiments that exceed the scope of this paper. Besides the observed differences for particular LLMs we also notice patterns across model families (e.g., the strong recovery of all DeepSeek-R1-Zero models with increasing $\rho$ vs the weaker recovery of the Qwen3 models). The emergence of reinforcement learning on verifiable rewards [Guo et al., 2025] presents a promising avenue to employ CogniLoad in the training process of LLMs, as the generated metadata of each experiment allows the precise verification of each reasoning step in light of the still scarce available training data of this type.

Despite these considerations, by decomposing the "task difficulty" into principled, controllable dimensions derived from cognitive science, CogniLoad provides a more insightful perspective than single-score benchmarks. It allows a more differentiated understanding of LLM reasoning capabilities and limitations, paving the way for more targeted development of robust and generalizable AI systems.

# 5    Conclusion

We introduced **CogniLoad**, a novel synthetic benchmark grounded in Cognitive Load Theory, for multi-dimensional evaluation of LLM long-context reasoning. By independently controlling parameters for intrinsic cognitive load ($d$), extraneous cognitive load ($\rho$), and task length ($N$ as a proxy for germane load demands), CogniLoad offers unprecedented diagnostic precision. Our evaluations revealed task length as a dominant performance constraint and uncovered unique "cognitive fingerprints" of LLM sensitivities to different load types, providing actionable insights beyond single-score benchmarks. CogniLoad offers a reproducible, scalable, and theoretically-grounded tool to systematically dissect LLM reasoning limitations and guide the development of more capable and robust AI systems. While human and artificial cognition are mechanistically distinct, applying frameworks like CLT to AI evaluation can provide valuable perspectives for understanding and characterizing their operational differences and capabilities.

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
