# Supplementary Material
## CogniLoad: A Synthetic Natural Language Reasoning Benchmark With Tunable Length, Intrinsic Difficulty, and Distractor Density

## 1 Investigating $\rho$ relative to $N$ and $d$

In Figure 1 of the paper we discover a characteristic U-shape of the performance of LLMs on CogniLoad relative to $\rho$ (i.e. the ratio of distractors to essential elements). The following figures investigate this U-shape at different levels of difficulty (Figure 1) and statement length (Figure 2).

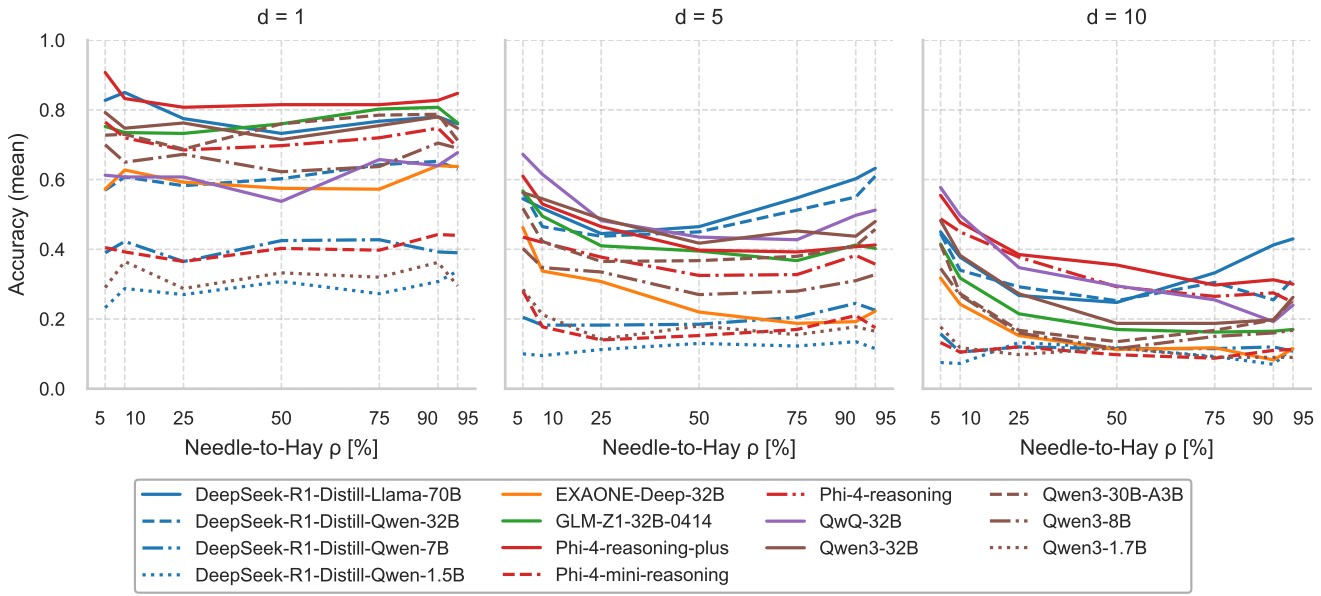

Figure 1: Distractor ratio $\rho$ relative to $d$ for $d \in \{1, 5, 10\}$

Figure 1 shows how average accuracy of the LLMs changes as the proportion of distractors ($\rho$, x-axis) grows under three difficulty settings $d = 1, 5, 10$. For the easiest tasks ($d = 1$, left panel) most models start high, drop slightly, and recover with a lower amount of distractors, producing the characteristic shallow U-shape reported in the paper. As difficulty rises to $d = 5$ (centre) the dip deepens: accuracy falls sharply between $\rho = 10\%$ and $25\%$, bottoming out near $50\%$ before some models—especially the larger DeepSeek and Qwen variants—begin to rebound above $\rho = 75\%$. Under the hardest condition ($d = 10$, right) most LLMs never recover; accuracy plateaus below 0.3 even when distractors are minimal. The plots confirm that (i) the U-shape flattens and shifts downward with higher $d$; (ii) model size and training recipe drive the height of both peaks; and (iii) the "valley" around moderate $\rho$ values is the most adversarial regime, where reasoning over a balanced mixture of signal and noise remains most difficult.

Figure 2 illustrates how the U-shaped accuracy pattern evolves as the total statement length $N$ increases from

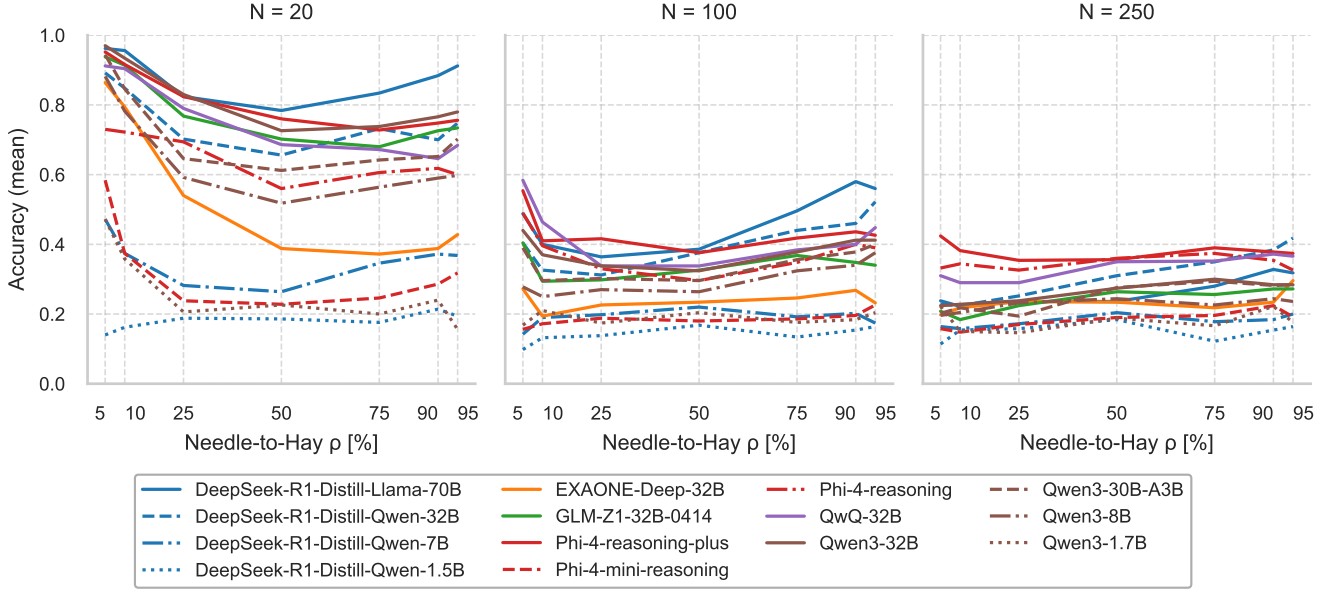

Figure 2: Distractor ratio $\rho$ relative to $N$ for $N \in \{20, 100, 250\}$

20 (left) to 100 (centre) and 250 (right). When $N = 20$ the curve is steep: accuracy starts near 0.9, plunges to its relatively lowest point around $\rho \approx 50\%$, and then partially rebounds once needles dominate. At $N = 100$ the same qualitative shape remains but is compressed: peak accuracy is lower and the minimum is reached sooner while most models still demonstrate a recovery at increasing $\rho$. By $N = 250$ the U has almost flattened and display only a modest uptick. It appears longer puzzles systematically erode both wings of the U—shaped precision on high-noise prompts and late recovery under low-noise prompts.

These plots seem to highlight a separation of the models into 2 classes, this is particularly visible for $d = 1$, $d = 5$ and $N = 20$. There is a gap in the accuracy between the group of smallest models and the largest. EXAONE being an exception as a large 32B model which is joining the group of the small models for certain configurations.

## 2 Model performance with Error-Bars

Due the small size and the large amount of models we compare in Figure 1 of the main paper we omit error-bars to improve legibility. In this section we plot larger versions of the charts in the panel with vertical error bars representing 90% confidence intervals for the mean accuracy of each model–condition pair.

We compute these confidence intervals with the Wilson score method for a binomial proportion. Specifically, for a given point with $k$ correct answers out of $n$ trials we set $\hat{p} = k/n$ and use the 90 % standard-normal quantile $z = 1.644853627$ to obtain

$$[l, \ u] \ = \ \frac{\hat{p} + \frac{z^2}{2n} \ \pm \ z\sqrt{\dfrac{\hat{p}\,(1 - \hat{p}) + z^2/(4n)}{n}}}{1 + \frac{z^2}{n}}.$$

The resulting interval $[l, u]$ marks the range that would contain the true underlying accuracy in 90% of repeated experiments with the same sample size. Wider bars correspond to greater sampling uncertainty, whereas tighter bars indicate more stable estimates.

The narrow error bars we observe substantiate the discussion of the results in the main paper and highlights a significant difference between the curves at 90% significance and the presence of the characteristic U-shape.

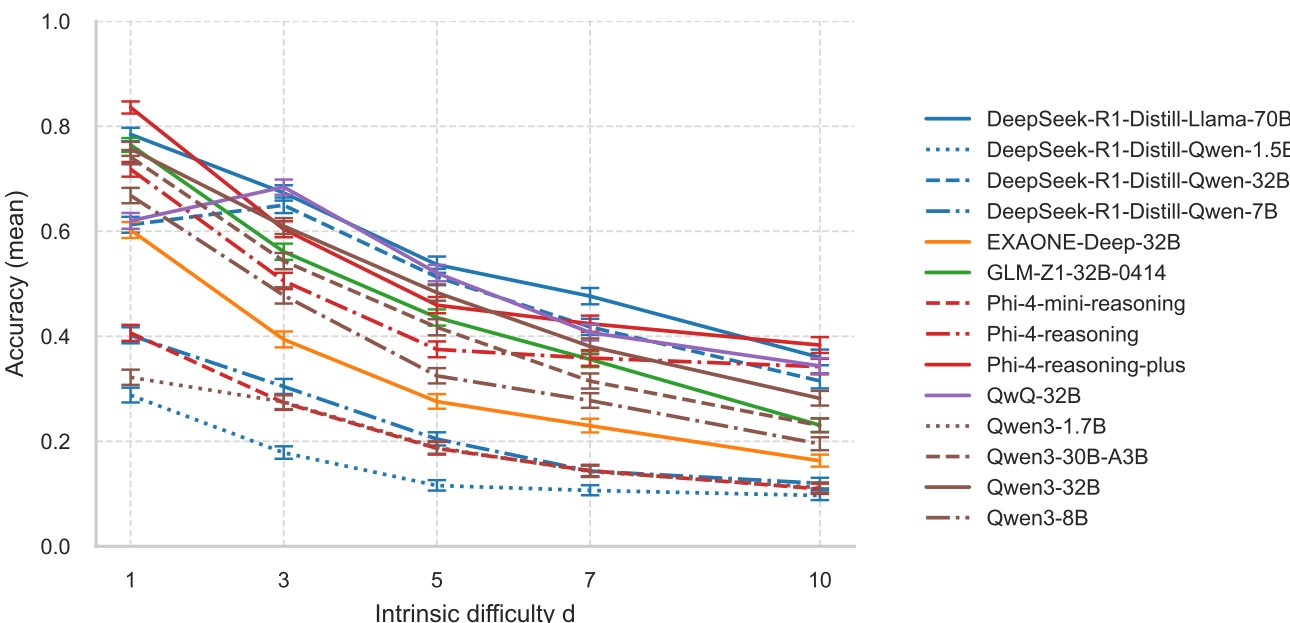

Figure 3: Accuracy relative to difficulty $d$ for $d \in \{1, 3, 5, 7, 10\}$ on the X-Axis

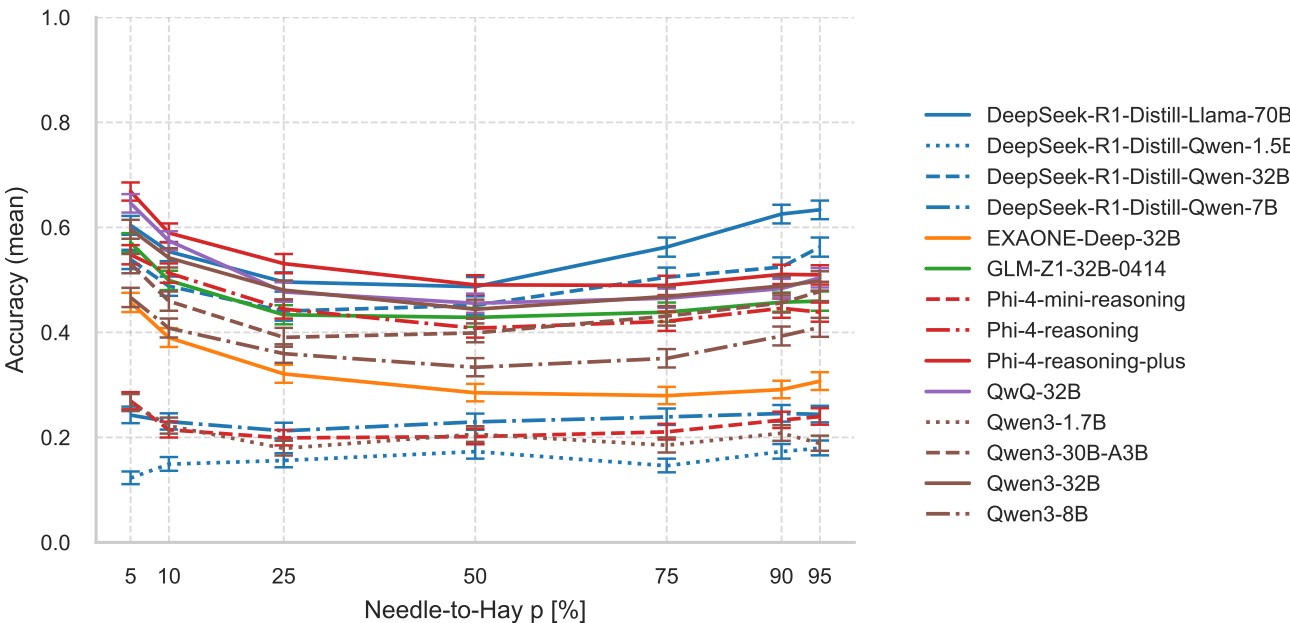

Figure 4: Accuracy relative to distractor-ratio $\rho$ for $\rho \in \{5, 10, 25, 50, 75, 90, 95\}$ on the X-Axis

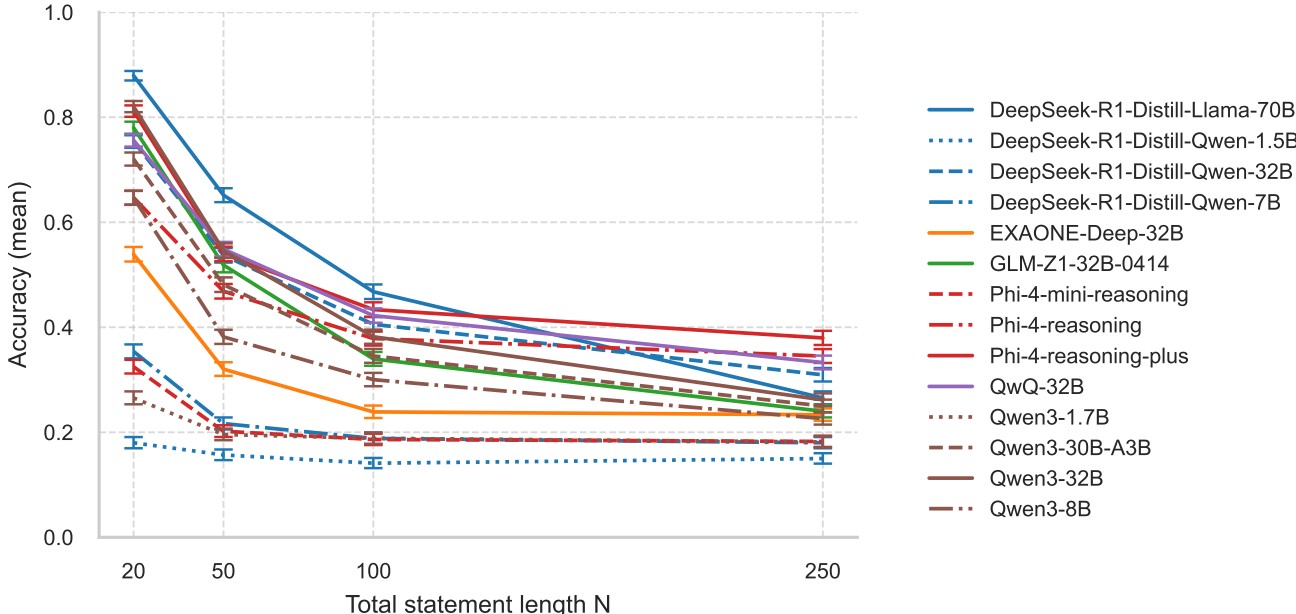

Figure 5: Accuracy relative to total statement length $N$ for $N \in \{20, 50, 100, 250\}$ on the X-Axis

# 3   AIC-Comparison

In this section we compare the Akaike Information Criterion (AIC) of the GLM model containing a pure linear $\rho$ term with a GLM model with a squared term added for $\rho$. The equations for the models are as following with $Y = 1$ indicating a correctly solved puzzle:

**Linear model**
$$\Pr\!\big(Y{=}1\big) = \sigma\big(\beta_0 + \beta_d\,d + \beta_N\,\log_{10}N + \beta_\rho\,\rho\big),$$

**Quadratic model**
$$\Pr\!\big(Y{=}1\big) = \sigma\big(\beta_0 + \beta_d\,d + \beta_N\,\log_{10}N + \beta_\rho\,\rho + \beta_{\rho^2}\,\rho^2\big),$$

To assess whether the quadratic specification provides a statistically significant improvement over the linear GLM, we compare their maximised log-likelihoods $\ell_{\text{lin}}$ and $\ell_{\text{quad}}$. The likelihood-ratio statistic $D = 2(\ell_{\text{quad}} - \ell_{\text{lin}})$ follows, under the null hypothesis that the extra term is unnecessary, a chi-squared distribution with one degree of freedom because the quadratic model introduces exactly one additional parameter.

The $p$-value reported in Table 1 is the upper-tail probability $p = \Pr(\chi_1^2 \geq D)$. p-values below 0.05 indicate that the quadratic term yields a statistically significant gain in fit and therefore justifies its inclusion.

We find that for all except for two models (i.e. DeepSeek-R1-Distill-Qwen-1.5B and DeepSeek-R1-Distill-Qwen-7B), the quadratic term for $\rho$ results in a significantly improved AIC value. Therefore we include the quadratic $\rho$ term in the GLM specification within the paper.

# 4   Complete Attribute Ontology

The following section lists the full attribute ontology of all values available for the categories.

| model | $p_{\text{LR}}$ | $\text{AIC}_{\text{linear}}$ | $\text{AIC}_{\text{quad}}$ |
|---|---|---|---|
| DeepSeek-R1-Distill-Llama-70B | **0.000** | 14348.157 | **14166.318** |
| DeepSeek-R1-Distill-Qwen-1.5B | 0.197 | 11719.491 | 11719.825 |
| DeepSeek-R1-Distill-Qwen-32B | **0.000** | 16951.857 | **16875.360** |
| DeepSeek-R1-Distill-Qwen-7B | 0.054 | 14121.021 | 14119.304 |
| EXAONE-Deep-32B | **0.000** | 15366.722 | **15277.989** |
| GLM-Z1-32B-0414 | **0.000** | 14635.661 | **14552.099** |
| Phi-4-mini-reasoning | **0.000** | 13833.347 | **13801.739** |
| Phi-4-reasoning | **0.000** | 17629.347 | **17568.414** |
| Phi-4-reasoning-plus | **0.000** | 16287.986 | **16203.301** |
| QwQ-32B | **0.000** | 17002.537 | **16870.080** |
| Qwen3-1.7B | **0.000** | 13692.272 | **13677.646** |
| Qwen3-30B-A3B | **0.000** | 15429.289 | **15314.110** |
| Qwen3-32B | **0.000** | 14902.252 | **14803.817** |
| Qwen3-8B | **0.000** | 15563.013 | **15457.894** |

Table 1: Model comparison: linear vs. quadratic fit. A bold number indicates a significant improvement of $\text{AIC}_{\text{quad}}$ over $\text{AIC}_{\text{linear}}$ with $p < 0.05$.

**people**  Peter, Paul, Mary, John, Mark, Jeff, Craig, Daniel, Anna, Arnoldo, Ali, Benjamin, Joe, Donald, Mitch, Chuck, Jack, Lucas, Jeniffer, Adam, Greg, Allan, David, Ellen, Fred, Hank, Hubert, Ian, Ingrid, Rebecca, Ken, Lewis, Michael, Nathaniel, Oliver, Russ, Steve, Sandy, Ted, Tanya, Veronica, Vincent, Wesley, Brad, Sam, Igor, Sue, Jan, Jeffrey, Jacques, Debby, Olivia, Benedict, Chris, Charles, Harry, Eli, Mahmoud, Chen, William, Linda, Elizabeth, Robert, Jennifer, Emily, Joseph, Thomas, Patricia, Anthony, Jessica, Brian, Lisa, Kevin, Karen, Laura, Eric, Stephanie, Michelle, George, Andrew, Joshua, Amber, Timothy, Victoria, Richard, Cynthia, Brandon, Megan, Matthew, Nancy, Jacqueline, Gary, Dorothy, Edward, Kimberly, Scott, Sara, Justin, Brittany, Ronald, Deborah, Janet, Christopher, Alexander, Samantha, Oscar, Cindy, Frank, Carl, Paula, Irene, Theresa, Dennis, Ralph, Gerald, Martin, Terry, Bryan, Lance, Corey, Casey, Brent, Derek, Travis, Austin, Victor, Jesse, Zachary, Kyle, Aaron, Betty, Connie, Holly, Donna, Gloria, Carla, Isabel, Sylvia, Evelyn, Doris, Arthur, Raymond, Harold, Lawrence, Neil, Brenda, Tracy, Simon, Wendy, Zoe, Ethan, Calvin, Sean, Ruth, Sheila, Miriam, Lorraine, Fay, Sophie

**clothes_socks**  blue, red, yellow, green, purple, pink, orange, black, white, gray

**clothes_shirt**  blue, red, yellow, green, purple, pink, orange, black, white, gray

**clothes_pant**  blue, red, yellow, green, purple, pink, orange, black, white, gray

**clothes_hat**  blue, red, yellow, green, purple, pink, orange, black, white, gray

**clothes_gloves**  blue, red, yellow, green, purple, pink, orange, black, white, gray

**clothes_underwear**  blue, red, yellow, green, purple, pink, orange, black, white, gray

**hair**  blue, red, yellow, green, purple, pink, orange, black, white, gray

**recent_eat**  pizza pasta burrito sushi taco burger toast egg banana potatoes

**recent_listen**  rock, pop, country, electronic, folk, jazz, blues, classical, funk, ska, rap, synth, disco, reaggea

**recent_watch**   drama, comedy, thriller, romance, adventure, horror, sci-fi, action, western, fantasy, documentary, mystery, crime, musical

**recent_read**   fiction, mystery, novel, thriller, biography, sci-fi, non-fiction, essay, encyclopedia, dictionary

**location**   bathroom livingroom kitchen basement toilet balcony garden pool bedroom store university farm office bank tree museum school airport zoo train bus park butcher library restaurant mall mountain tunnel church river pond harbor taxi gallery bar pizzeria beach gym elevator insurance embassy police hospital festival monument laboratory observatory valley motorway viewpoint synagogue factory castle cave stadium arena cabin plaza amphitheater bridge pier vineyard forest cliff desert creek bay lighthouse orchard resort camp inn motel aquarium bazaar chapel monastery lookout campground retreat dock depot consulate manor theatre cathedral casino lodge mill bakery spa station diner gazebo terrace arcade boardwalk winery hill plateau ridge port oasis market fairground quarry mine grove auditorium cemetery dunes courthouse prison fort granary ranch promenade coliseum field tower pavilion silo bistro labyrinth cafe saloon brewery carnival marina estate safari cottage courtyard waterpark island greenhouse meadow lagoon ford hacienda village marketplace grotto maze golfcourse atrium academy waterfront peninsula cove summit plains