# OpenReview forum: "CogniLoad: A Synthetic Natural Language Reasoning Benchmark With Tunable Length, Intrinsic Difficulty, and Distractor Density"
_NeurIPS.cc/2025/Datasets_and_Benchmarks_Track — Submitted to NeurIPS 2025 Datasets and Benchmarks Track_

### Official Review · Reviewer_UGjY · 2025-06-14

**Rating:** 4
**Confidence:** 4

**Summary:**

This paper introduces CogniLoad, a novel synthetic benchmark grounded in Cognitive Load Theory (CLT), which generates natural-language logic puzzles with independently tunable parameters:  intrinsic difficulty, distractor-to-signal ratio and task length. They conduct extensive experiments reveal distinct performance sensitivities, identifying task length as a dominant constraint and uncovering varied tolerances to intrinsic complexity and U-shaped responses to distractor ratios.

Their contribution can be summarized as: (1). The collection of CogniLoad, a tunable length, intrinsic difficulty, and distractor density dataset for the reasoning benchmark. (2). An algorithm for the automatic randomized generation and evaluation of puzzle instance. (3). Empirical results and insights on 14 SOTA reasoning LLMs.

**Dataset Code Accessibility:**

Yes

**Ethical Considerations:**

No, there are no or only very minor ethics concerns

**Final Justification:**

The authors addressed one of my concerns in terms of generalization by conducting more experiments with powerful and closed-sourced LLMs. Therefore, I keep my positive attitude towards this paper.

**Limitations Weaknesses:**

* While the authors claimed they use 14 SOTA LLMs in the experiment section, there are only open-source models involved. Powerful closed-source models like DeepSeek-R1, OpenAI-o3 and gemini-2.5-pro are not included. The lack of these real SOTA LRMs weakens the generalization of the findings and insights in this paper.

* In this paper, the definition of puzzles involves a series of statements updating the attributes according to conditions specified in each statement. This definition, compared with previous works in puzzle datasets for LLMs benchmark, is relatively limited [1]. The conclusion and findings drawn from these single-type questions might not be generalized enough to reflect the general mechanism of LLMs' reasoning.

[1] Giadikiaroglou, Panagiotis, et al. "Puzzle solving using reasoning of large language models: A survey." arXiv preprint arXiv:2402.11291 (2024).

**Strengths Contributions:**

* This paper is well-written, with clear tables and figures, making it easy to follow and understand.

* The collected CogniLoad dataset is potentially useful for the community to benchmark and understand LLMs' performance under various factors.

* The provided insights could be helpful for future work in LRMs to boost foundational puzzle reasoning capability.

---

> ### Author Rebuttal · Authors · 2025-07-30
>
> Thank you for your clear questions and criticism which we are answering in turn below, your text in italics.
>
> ### *Limitation 1: While the authors claimed they use 14 SOTA LLMs in the experiment section, there are only open-source models involved. Powerful closed-source models like DeepSeek-R1, OpenAI-o3 and gemini-2.5-pro are not included. The lack of these real SOTA LRMs weakens the generalization of the findings and insights in this paper.*
>
> Thank you for this suggestion. Indeed, it is a very legitimate point made by 3 out of the 4 reviewers, namely Reviewer CZW5, fFRQ and UGjY. Please read our answer to limitation 1 of Reviewer fFRQ, where we provide results for these frontier models (i.e., o4-mini, gemini-2.5-pro, gemini-2.5-flash, gemini 2.5.flash-lite, and DeepSeek-R1-0528).
>
> ### *Limitation 2: In this paper, the definition of puzzles involves a series of statements updating the attributes according to conditions specified in each statement. This definition, compared with previous works in puzzle datasets for LLMs benchmark, is relatively limited [1]. The conclusion and findings drawn from these single-type questions might not be generalized enough to reflect the general mechanism of LLMs' reasoning.[1] Giadikiaroglou, Panagiotis, et al. "Puzzle solving using reasoning of large language models: A survey." arXiv preprint arXiv:2402.11291 (2024).*
>
> We thank the reviewer for highlighting the comprehensive survey by Giadikiaroglou et al. [1]. This is a crucial point that gets to the heart of our benchmark's design philosophy.
>
> The reviewer correctly observes that CogniLoad employs a single, specific type of logic puzzle, which is indeed more limited in scope than the broad taxonomy of puzzles (e.g., riddles, Sudoku, chess, commonsense problems) presented in the survey. We agree with this characterization, and in fact, this is a deliberate design choice of our benchmark, aimed at achieving a different, more diagnostic goal than general-purpose puzzle benchmarks.
>
> The primary motivation for CogniLoad, as detailed in our introduction, is to address a key shortcoming in existing benchmarks: they often "blur critical factors like intrinsic task complexity, distractor interference, and task length." The impressive scope of tasks surveyed in [1] exemplifies this challenge. While these benchmarks are excellent for assessing an LLM's general, multifaceted reasoning ability, their heterogeneity makes it nearly impossible to isolate the specific cause of a failure. For example, if a model fails on a benchmark that mixes riddles and logic puzzles, is it due to a deficit in multi-step deduction, an inability to handle a long context, or susceptibility to distractor information? The cause remains ambiguous.
>
> By using a single, parameterized puzzle generator, CogniLoad can independently vary the parameters that operationalize Cognitive Load Theory. This factorial control is what enables the precise, diagnostic insights that are the central contribution of our work. It allows us to move beyond a single accuracy score and create "cognitive fingerprints" for each model, revealing how its performance changes as we stress each cognitive dimension independently. We can determine if a model's long-context failure stems from the sheer length (N), the complexity of the core task (d), or the density of irrelevant information (p). Our findings, such as the distinct U-shaped response to distractor density or the identification of task length as the most dominant constraint, are only possible because we have controlled for other confounding variables by keeping the core task type constant.
>
> We see CogniLoad as a complementary tool to the broader benchmarks. While the benchmarks surveyed in [1] test for generality, CogniLoad tests for robustness under specific, quantifiable stressors. It is designed to be a "stress test" or a diagnostic instrument for understanding the underlying mechanics of LLM reasoning, rather than a single measure of its overall puzzle-solving prowess.
>
> Regarding the reviewer's valid concern about generalizability, we acknowledge this in Section 4 ("Nuances of the CLT-LLM Analogy" and "Scope of Reasoning and Generalizability"). Our goal is not to claim that performance on CogniLoad's logic puzzles will perfectly predict reasoning performance on riddles. Instead, we propose that the sensitivities to intrinsic complexity, distractor interference, and sustained reasoning are fundamental variables that influence reasoning on tasks by LLMs and are likely to have implications across a range of reasoning tasks.
>
> To make this crucial point clearer to the reader, will explicitly contrast our approach with the "general-purpose" survey-style benchmarks in our paper, better articulating that the "limited" scope of CogniLoad is a necessary trade-off for achieving diagnostic precision and factorial control. We believe this clarification will help frame our contribution more accurately as a diagnostic tool that complements, rather than competes with, benchmarks that aim for breadth.

---

> > ### Comment · Reviewer_UGjY · 2025-08-04
> >
> > Thanks for the response, it addressed one of my concerns in terms of generalization by conducting more experiments with powerful and closed-sourced LLMs. Therefore, I will keep my positive attitude towards this paper. Please add all the added content to the camera-ready version to make this work more solid.

---

> > > ### Author Response · Authors · 2025-08-05
> > >
> > > Dear Reviewer UGjY,
> > >
> > > Thank you for your efforts and supporting our work. In the camera ready, we will add the results of the proprietary models, cite (Giadikiaroglou, et al., 2024), and better clarify our contributions in the context of previous work.

---

> ### Comment · Area_Chair_Sy7t · 2025-08-04
> **Please engage with authors during the discussion period**
>
> Dear reviewer UGjY,
>
> Thank you for posting the acknowledgment of the authors' rebuttal. Please note that the acknowledgment you added includes the phrase "I have engaged in discussions and responded to authors." An important part of the discussion period is the opportunity to discuss with the authors whether their rebuttal has addressed your concerns or, if not, what information the authors should provide or what changes they should make to address them, to the degree that that is possible. Please add a response to the rebuttal to indicate this. There are only a couple of days left in the discussion period, and your engagement during this time is critical.

---

### Official Review · Reviewer_fFRQ · 2025-07-01

**Rating:** 4
**Confidence:** 4

**Summary:**

The paper introduces CogniLoad, a synthetic natural-language logic-puzzle benchmark that factors long-context reasoning difficulty into three cognitively motivated axes: intrinsic difficulty d, task length N (a proxy for germane load), and distractor density ρ (extraneous load). A controlled generator produces puzzles whose parameters can be varied independently; exact-match accuracy is used as the evaluation metric. The authors evaluate 14 open-weights “reasoning” LLMs on 14 000 puzzle instances each, then fit a binomial GLM to derive interpretable capacity points (ECL50, NT50, ID50) that capture model-specific sensitivities. Results reveal, among other findings, that longer chains of updates are the dominant failure mode, while distractors produce a U-shaped accuracy curve.

**Dataset Code Accessibility:**

Yes

**Ethical Considerations:**

No, there are no or only very minor ethics concerns

**Final Justification:**

Given the authors' response, I raised my score by 1.

**Limitations Weaknesses:**

Missing evaluation of frontier proprietary models – The study restricts itself to open-weights LLMs; consequently, the newest  GPT O-series models (O3, O3-mini, O4-mini) or Gemini 2.5 —arguably the state-of-the-art for long-context reasoning—are absent. This makes it hard to tell whether the trends revealed by CogniLoad generalize to today’s best-performing closed models.

The presentation results are hard to follow. For example, Table 3 packs 10 numeric columns of GLM coefficients plus three derived thresholds into a single wide block (p-values and asterisks), with no visual cues to highlight key take-aways. Readers must manually scan rows to see which models excel at ECL50 vs. ID50.

 Although natural-language templates are used, puzzles may lack linguistic diversity and pragmatic cues found in real documents; this could limit ecological validity.

**Strengths Contributions:**

The experiments are comprehensive. 4 diverse open-weight LLMs are evaluated over 140 configurations; Figure 1 and the regression analysis give fine-grained performance maps and interpretable thresholds.

Using GLM offers interpretable results. It reveals task length (N) as the dominant bottleneck and quantifies distractor robustness via NT50; this goes beyond raw accuracy tables.

---

> ### Author Rebuttal · Authors · 2025-07-30
>
> Thank you for your review and criticism which we are answering in turn below, your text in italics.
>
> ### *Limitation 1: Missing evaluation of frontier proprietary models – The study restricts itself to open-weights LLMs; consequently, the newest GPT O-series models (O3, O3-mini, O4-mini) or Gemini 2.5 —arguably the state-of-the-art for long-context reasoning—are absent. This makes it hard to tell whether the trends revealed by CogniLoad generalize to today’s best-performing closed models.*
>
> This is a point raised by three Reviewers: CZW5, fFRQ and UGjY so we provide a comprehensive answer here.
>
> We recognize it is a very valid concern from the reviewers, as it could be the case that the frontier proprietary models are so powerful that they saturate the benchmark already, and as a consequence our benchmark would lose its relevance. We also recognize that providing benchmark results for proprietary LLMs will increase the relevance of our work to the general public and thereby increase its impact.
>
> For these reasons we have evaluated CogniLoad on the 5 leading proprietary models: OpenAI o4-mini, Deepseek R1-0528 and  Gemini-2.5-pro, Gemini-2.5-flash, and Gemini-2.5.flash-lite by Google. The average accuracies (computed similar to Figure 1) are shown below. Due to rate-limit limitations by OpenAI we were unable to evaluate o3 while time constraints of this response limited the evaluation to a representative subset of 10 puzzles for each parameter configuration (i.e. a total of 1’400 puzzle instances). The revised camera ready version of the paper affords the time to also include o3 and a full evaluation based on all 14’000 instances.
>
>
> ### Intrinsic difficulty (d)
> | |1  |3  |5  |7  |10 |
> |---------------------|---|---|---|---|---|
> |o4-mini-2025-04-16   |88%|80%|78%|70%|64%|
> |gemini-2.5-pro       |94%|74%|72%|69%|65%|
> |gemini-2.5-flash     |81%|58%|51%|48%|48%|
> |gemini-2.5-flash-lite|70%|48%|36%|22%|17%|
> |DeepSeek-R1-0528      |88%|80%|68%|57%|48%|
>
>
>
> ### Task length (N)
> | |20 |50 |100|250|
> |---------------------|---|---|---|---|
> |o4-mini-2025-04-16   |98%|92%|75%|38%|
> |gemini-2.5-pro       |99%|98%|82%|21%|
> |gemini-2.5-flash     |84%|79%|53%|12%|
> |gemini-2.5-flash-lite|56%|35%|34%|30%|
> |DeepSeek-R1-0528      |97%|87%|58%|29%|
>
>
>
> ### Distractor ratio (p)
> | |5% |10%|25%|50%|75%|90%|95%|
> |---------------------|---|---|---|---|---|---|---|
> |o4-mini-2025-04-16   |82%|82%|73%|67%|72%|77%|81%|
> |gemini-2.5-pro       |77%|76%|74%|73%|75%|78%|74%|
> |gemini-2.5-flash     |59%|61%|57%|52%|56%|58%|57%|
> |gemini-2.5-flash-lite|54%|47%|40%|32%|29%|32%|39%|
> |DeepSeek-R1-0528      |74%|71%|65%|64%|64%|67%|74%|
>
>
> ### Capacity points at 50% accuracy
> | model | ECL50 | NT50 | ID50 |
> |---|---|---|---|
> | o4-mini-2025-04-16 | 130.6 | -- | 10.61 |
> | gemini-2.5-pro | 152.1 | -- | 12.74 |
> | gemini-2.5-flash | 81.0 | -- | 6.26 |
> | gemini-2.5-flash-lite | 9.5 | 0.92 | 1.8 |
> | DeepSeek-R1-0528 | 110.7 | -- | 7.77 |
>
> The new results indicate that the closed-source commercial models are slightly better than the open-source ones, but not always. Our benchmark is not saturated. We see the same decaying curves in the task length and in the intrinsic difficulty, as for the open-sources models. Particularly interesting is that the characteristic U-shaped response to distractor ratios is equally present for the proprietary frontier models. Also noteworthy is that at the maximum task length (250) gemini-flash-lite beats gemini-flash and gemini-pro which appears to be a result of higher reasoning efficiency within the limited context. Please see our response to reviewer CWZ5 (Additional feedback 2) for empiric data for this observation.
>
> ### *Limitation 2: The presentation results are hard to follow. For example, Table 3 packs 10 numeric columns of GLM coefficients plus three derived thresholds into a single wide block (p-values and asterisks), with no visual cues to highlight key take-aways. Readers must manually scan rows to see which models excel at ECL50 vs. ID50.*
>
> Thank you for the feedback. We will improve the legibility of Table 3 with the extra page the camera-ready version of the paper offers. We agree that our scores, which are central for our evaluation, are not highlighted enough in this big and compressed table. We will make a new table to make them clearer and clearly highlight the top scores achieved by each of the LLMs to make comparison easy for the reader.
>
> ### *Limitation 3: Although natural-language templates are used, puzzles may lack linguistic diversity and pragmatic cues found in real documents; this could limit ecological validity.*
>
> On linguistic diversity:
>
> This is a valid and very interesting point. For example, we could have designed the puzzle to generate the text in different languages. It is possible that LLMs trained on more Chinese texts may perform better if the puzzle text is in Chinese. Recognizing that the reasoning performance depends on the linguistic diversity would be an interesting discovery. This ability to generate puzzles automatically, following the same deductive reasoning pattern while allowing to vary the diversity of the text can be seen as an additional strength of our benchmark that is still unexploited and presents a promising avenue for future extensions. Please see also answer to Reviewer CWZ5 (additional feedback 1).
>
> Regarding pragmatic cues in real documents:
>
> If we interpret your remark correctly, then these pragmatic cues would simplify finding the solution to the puzzle by bypassing the necessity to reason through some statements. While it would be an interesting question to systematically study if LLMs can take these shortcuts when available, we purposefully designed CogniLoad so that no shortcuts can be taken and the LLM rigorously needs to reason through every statement of the puzzle. We decided to do this because in the context of benchmarks, these shortcuts are usually seen more as a problem than as a feature. A more specific version of CogniLoad in the future could however incorporate such on purpose to really measure how good different LLMs are at utilizing them.

---

> > ### Comment · Reviewer_fFRQ · 2025-08-08
> >
> > Thanks for the response. I raise my score based on the new experiment result.

---

> ### Comment · Area_Chair_Sy7t · 2025-08-04
> **Please engage with authors during the discussion period**
>
> Dear reviewer fFRQ,
>
> The authors have responded to your review with a rebuttal to address the three limitations that you noted. Please review the rebuttal that they have posted and respond to indicate whether your concerns have been addressed, or, if not, what information the authors should provide or what changes they should make to address them, to the degree that that is possible. There are only a couple of days left in the discussion period, and your engagement during this time is critical.

---

> ### Author Response · Authors · 2025-08-06
>
> Dear reviewer fFRQ,
>
> We are currently unaware of the extent to which our responses have addressed and clarified your comments, and willing to provide further clarifications. Do you have any remaining concerns? If we have sufficiently addressed your concerns, we hope you could re-evaluate our work more positively.

---

> ### Comment · Area_Chair_Sy7t · 2025-08-07
> **[second reminder] please engage with authors**
>
> Dear reviewer fFRQ,
>
> Please let the authors know if they've addressed your concerns. It doesn't have to be a long message, but they should hear from you about whether you found their rebuttal satisfactory or if you have remaining questions. The discussion period closes in two days, and they have not yet heard from you.

---

> ### Author Response · Authors · 2025-08-09
>
> Dear Reviewer fFRQ,
>
> Thank you for your efforts and review of our paper! We are glad that we could address your concerns and grateful you raised raised your score.

---

### Official Review · Reviewer_tRiH · 2025-07-04

**Rating:** 4
**Confidence:** 3

**Summary:**

The submission grounds evaluating the reasoning abilities of LLMs in Cognitive Load Theory (CLT), which is a theory of human cognition that decomposes the cognitive load into three dimensions: intrinsic cognitive load (ICL), extraneous cognitive load (ECL), and germane cognitive load (GCL).

Specifically, the submission designs CogniLoad, a natural-language state-tracking puzzle synthesis program that resembles

1. ICL via intrinsic puzzle difficulty ($d$), i.e., the maximum number of conditions and state updates in the statements,
2. ECL via distractor density ($\rho$), i.e., the ratio of statements unrelated to the instance of interest,
3. GCL via task length ($N$), i.e., the total number of statements.

These parameters are all tunable, enabling systematic evaluation.

Based on this puzzle synthesis program, the submission evaluates 14 reasoning LLMs and shows some interesting results.

**Additional Feedback:**

1. It might be interesting to investigate whether models can hack such synthetic puzzles by writing programs.

**Dataset Code Accessibility:**

Yes

**Dataset Code Comments:**

The submission provides the code link.

**Ethical Considerations:**

No, there are no or only very minor ethics concerns

**Final Justification:**

The rebuttal has basically resolved my concerns.

**Limitations Weaknesses:**

1. The idea of synthesizing puzzles with multi-dimensional controllable parameters is not novel. There are similar previous works like GSM-Infinite.
2. There lacks insightful analysis of the scaling curves of parameters that can help improve the modelling.
3. The analogy between human cognition and LLM reasoning might be nuanced (which is mentioned in the paper).

**Strengths Contributions:**

1. Generally, the paper is well-written, organized and easy to understand, and the figures, tables, and captions are informative.
2. The submission provides a new puzzle synthesis programs with three tunable parameters, which might be useful.

---

> ### Author Rebuttal · Authors · 2025-07-30
>
> Thank you for your clear questions and criticism which we are answering in turn below, your text in italics.
>
> ### *Limitation 1: The idea of synthesizing puzzles with multi-dimensional controllable parameters is not novel. There are similar previous works like GSM-Infinite.*
>
> We sincerely thank the reviewer for highlighting the important comparison with GSM-Infinite which we will cite as a related work. Due to the similarity of this benchmark and CogniLoad we appreciate the opportunity to highlight the key differences that illustrate how our novelty lies not in the fact that there are multi-dimensional controllable parameters but that these parameters can be adjusted _independently_ (as claimed as a contribution in our submission).
>
> The parameters of GSM-Infinite’s are inherently coupled, to increase its primary metric of reasoning complexity (i.e. the number of operations) more nodes and edges need to be added to the underlying computational graph, and to add noise, extra non-essential nodes are added to the graph. Each new operation translates into additional sentences in the problem statement which automatically increases the puzzle length. Therefore it is impossible to test a problem with high difficulty or high noise in a short context as both variables necessarily increase the experiment length. Noise and difficulty are not independent parameters to the task length, but introduce extra components to the graph, automatically increasing the puzzles length.
>
> In contrast, CogniLoad treats these parameters as truly independent. It is possible to increase the difficulty without adding extra puzzle sequence steps, and it is possible to change the ratio of noise independently of the length of the puzzle. This feature of CogniLoad is why, to the best of our knowledge, it is the first benchmark to discover the characteristic U-shaped response to distractors that neither GSM-Infinite nor any of the benchmarks we initially reviewed demonstrate.
>
> Further, the benchmarks differ fundamentally in what they are designed to achieve. GSM-Infinite tries to measure the raw scaling limits (of sequential arithmetic reasoning in grade school math problems), while CogniLoad through its independent parameters aims to provide diagnostics about the cause of reasoning failures. It yields a detailed “cognitive fingerprint” for each LLM that highlights their relative strengths while pinpointing to the cause of why reasoning fails in the 3 dimensions of the cognitive load.
>
> GSM-Infinite requires basic mathematical knowledge for its arithmetic operations, while CogniLoad poses reasoning problems that can be answered with pure deduction without being a test of quantitative skills. This means while an LLM pretrained on more math datasets will perform better on GSM-Infinite, no specific pre-training of an LLM on domain knowledge will change its performance on CogniLoad, adding to the longevity of the benchmark.
>
> If you had the impression that CogniLoad lacks novelty compared to GSM-infinite, we hope our answer convinces you that this is not the case and hope very much that you want to consider updating your score to recommend accepting our submission.
>
> ### *Limitation 2: There lacks insightful analysis of the scaling curves of parameters that can help improve the modelling.*
>
> We are not sure how to interpret your remark. If it is about modelling the curves shown on Figure 1 (similar to the modelling done in GSM-infinite with the Sigmoid function), this is what we have done in section 3.1.1 with the generalized linear model, which goes beyond the GSM-infinite analysis by not only fitting the logistic function but also providing the parameters and using the modelled shape to define new capacity thresholds for reasoning, based on the curve shapes: ECL50, NT50 and ID50.
>
> If we interpret your point as being about investigating the failure modes, with the purpose to improve models: This is a very important and interesting aspect: We have made some initial checks on the reasoning process of a few models, and found different types of error (see our answer to Reviewer CWZ5, “Additional feedback”) with some empiric results. Is however beyond the scope and page limit of this paper to make such a systematic study. In this paper we focus on the benchmark design and a systematic performance evaluation with regards to tunable generation parameters as a first step. It is a difficult issue of course, because different failure modes point to multiple directions of possible improvement, and highly dependent on the underlying model architecture and its training recipe.
>
> ### *Limitation 3: The analogy between human cognition and LLM reasoning might be nuanced (which is mentioned in the paper).*
>
> We agree with and acknowledge the need to nuance this analogy. We have tried to do so in the paper as you also mention in parenthesis. More work is certainly needed to investigate this parallel further, in particular the study of the reasons of failure and success, which as mentioned will be the theme of a forthcoming paper. We however believe that it is a parallel worth illustrating as it is a connection that, to the best of our knowledge, no paper has explicitly highlighted. In the context of the difficult and increasingly relevant question of how human and computational cognition differ functionally, mapping the failure of LLMs directly to parameters that are established in cognitive psychology is a worthwhile perspective. In the revised version of the present paper we will try to discuss the analogy and its limit better.
>
> ### *Additional Feedback 1: It might be interesting to investigate whether models can hack such synthetic puzzles by writing programs.*
>
> This is indeed an interesting point to investigate. It is possible that an LLM could decide that it is more efficient to write a program to solve the puzzle rather than reasoning through the text. However, providing access to external tools would fundamentally undermine the purpose of a pure reasoning benchmark so we restrict such an access to truly investigate how well an LLM reasons and specifically how factors of the reasoning task influence its failure. Since the puzzles are sequential deductions by design, following all the puzzle steps is necessary to obtain the correct answer so writing a program to solve CogniLoad would be a shortcut that defeats this purpose.

---

> ### Comment · Area_Chair_Sy7t · 2025-08-04
> **Please engage with authors during the discussion period**
>
> Dear reviewer tRiH,
>
> The authors have responded to your review with a rebuttal to address the three limitations that you noted and responds to your other point of feedback. Please review the rebuttal that they have posted and respond to indicate whether your concerns have been addressed, or, if not, what information the authors should provide or what changes they should make to address them, to the degree that that is possible. There are only a couple of days left in the discussion period, and your engagement during this time is critical.

---

> ### Author Response · Authors · 2025-08-06
>
> Dear reviewer tRiH,
>
> We are currently unaware of the extent to which our responses have addressed and clarified your comments, and willing to provide further clarifications. Do you have any remaining concerns? If we have sufficiently addressed your concerns, we hope you could re-evaluate our work more positively.

---

> ### Comment · Area_Chair_Sy7t · 2025-08-07
> **[second reminder] please engage with authors**
>
> Dear reviewer tRiH,
>
> Please let the authors know if they've addressed your concerns. It doesn't have to be a long message, but they should hear from you about whether you found their rebuttal satisfactory or if you have remaining questions. The discussion period closes in two days, and they have not yet heard from you.

---

### Official Review · Reviewer_CwZ5 · 2025-07-05

**Rating:** 5
**Confidence:** 2

**Summary:**

This paper proposed CogniLoad, a synthetic benchmark evaluating long-context reasoning capabilities of LLMs grounded in Cognitive Load Theory. The benchmark is composed of natural-language logic puzzles with three independently tunable parameters. The authors generate thousands of puzzles and evaluate 14 state-of-the-art reasoning models. They analyze model performance by identifying distinctive sensitivities to each type of cognitive load and deriving interpretable thresholds for comparing the reasoning capabilities.

**Additional Feedback:**

I have some additional questions and comments:

- It is known that many models will be distrcted by small perturbation on the phrasing of the questions. How robust is the evaluation to small perturbations in puzzle phrasing?
- Is there any analysis of failure types? These may include error types such as distractor confusion vs. logic failure.
- I wonder how the attribute ontology is selected. For example, will expanding, changing or limiting the category name lead to different results? This contributed to the robustness of the benchmark.

**Dataset Code Accessibility:**

Yes

**Dataset Code Comments:**

The dataset can be easily accessed by load_dataset in Python.

**Ethical Comments:**

This is a synthetic dataset. I have no ethical concerns.

**Ethical Considerations:**

No, there are no or only very minor ethics concerns

**Final Justification:**

The work seems solid and promising after the rebuttal. The authors provide detailed explanations and largely addressed my concerns on the evaluation and the design of the dataset. So I will keep my positive score of 5.

**Limitations Weaknesses:**

The paper has following limitations and weaknesses:

- The benchmark is limited to sequential logic-grid puzzles. This restricts generalizability to other reasoning types like mathematics and code. I guess this may be solved by adding such elements into attribute ontology, but authors can provide furthur explainations.
- The evaluation is limited to 14 open-weight reasoning LLMs. It's unclear how well closed-source models (e.g., GPT-4, Claude)  and other open-weight LLMs (e.g., Llama) would perform and whether conclusions hold across other types of models (e.g., retrieval-augmented) since the tasks also need retrieval ability.
- The paper seems not to provide prompting format and decoding strategy used.

**Strengths Contributions:**

The paper has following strengths and contributions:

- The use of Cognitive Load Theory as a guiding principle is novel and well-motivated.
- The use of tunable parameters decomposes reasoning difficulty into interpretable, orthogonal components. This enabels precise manipulation of reasoning complexity, context length and some other factors, which is valuable for controlled studies.
- The derivation of ECL50, NT50, and ID50 offers clear and interpretable metrics for comparing model capacity.
- The dataset generation process is fully documented. Code and data are made publicly available in the proper formats.

---

> ### Author Rebuttal · Authors · 2025-07-30
>
> Thank you for your support and your thoughtful comments, which we will use to improve our paper. We follow your points, in order. Since 2 of your 3 concerns were also raised by other reviewers, who declared a higher confidence score than yours, we believe that your expertise in the area of our paper may be understated in your confidence score.
>
> ### *Limitation 1: The benchmark is limited to sequential logic-grid puzzles. This restricts generalizability to other reasoning types like mathematics and code. I guess this may be solved by adding such elements into attribute ontology, but authors can provide further explainations*
>
> We agree that our benchmark is not designed to test other more specialized types of reasoning and believe that is more of a strength than a weakness. The key benefit is that this formulation as a pure deductive reasoning task makes the performance of LLMs invariant towards domain knowledge in areas such as mathematics and code. LLMs cannot be simply pre-trained on more math problems or coding to improve their scores. Through this design, the evaluation of LLMs depends only on aspects of cognitive load that are also established in Cognitive Load Theory for humans. This point also relates to a similar point by Reviewer UGjY (point 2), please also see our answer to it for further details.
>
> Of course, as you correctly state, if more specialized areas of reasoning are of particular interest like coding or math, the ontology could be changed to cover these areas, and the state can represent intermediate variables in a program or series of mathematical operations. While these are more specialized versions of CogniLoad, in this paper we first want to introduce its most general form that can then be adapted.   Please see also our answer to point 3 raised by Reviewer fFRQ about linguistic diversity, which may be related to your point.
>
> ### *Limitation 2: The evaluation is limited to 14 open-weight reasoning LLMs. It's unclear how well closed-source models (e.g., GPT-4, Claude) and other open-weight LLMs (e.g., Llama) would perform and whether conclusions hold across other types of models (e.g., retrieval-augmented) since the tasks also need retrieval ability.*
>
> Thank you for this suggestion. Indeed, this is a very legitimate point made by 3 out of the 4 reviewers, namely Reviewer CZW5, fFRQ and UGjY. Please find our answer to limitation 1 of Reviewer fFRQ , where we provide results and discussion for frontier models (i.e., o4-mini, gemini-2.5-pro, gemini-2.5-flash, gemini 2.5.flash-lite, and DeepSeek-R1-0528).
>
> Since the focus of this benchmark is to evaluate LLMs that are trained to reason, GPT-4, Claude and Llama would not be appropriate comparisons. In early experiments we noticed that non-reasoning models fare extremely poorly at CogniLoad (failing already most of the times at problems with 5 steps) so we systematically excluded them in the evaluation.
>
> ### *Limitation 3: The paper seems not to provide prompting format and decoding strategy used.*
> Thank you for asking about this important aspect. We evaluated the models with the default decoding settings that the model developers specified in the model configurations, and only with the prompt as specified in figure 2 of the paper. We will list these default decoding settings explicitly in the paper appendix. The maximum context length per model is restricted to 32’768 tokens to evaluate each model at the same level of reasoning efficiency (we will highlight this information in the main text of the paper).
>
> ### *Additional Feedback 1: It is known that many models will be distracted by small perturbation on the phrasing of the questions. How robust is the evaluation to small perturbations in puzzle phrasing?*
> This is a good point. We opted for the most concise instruction that we kept constant for each example and asked a simple question about the target attribute. The randomness in the selection of the target attribute in the final question, in the statement generation, and in the used ontology attributes, introduce perturbations by design. This question may be also related to limitation 3 raised by fFRQ (diversity in the texts): please also see this answer to reviewer fFRQ.
>
> This question is important and we think it should be addressed separately. Revealing and understanding differences in LLM accuracy due to perturbation, change of vocabulary or language can be an interesting new measure of robustness and also an evaluation of fairness (with respect to language). It could lead to new insights on the inner reasoning mechanism of LLMs. In the present work we prefer to focus on setting a strong base for the evaluation, with the 3 dimensions of cognitive load and their analysis and leave this question for future work.
>
> ### *Additional Feedback 2: Is there any analysis of failure types? These may include error types such as distractor confusion vs. logic failure.*
>
> This is a very important and interesting aspect. We have made some initial checks on the reasoning process of some models, examining the thinking traces and found different types of error. Besides the expected cases of simple logic errors, or insufficiently efficient reasoning efficiency (within the context limit), one striking example is when the LLM loses track of the reasoning and goes into an infinite reasoning loop repeating to reason from the beginning of the problem until it reaches the context limit. Another notable example is LLMs starting to write incoherent sentences and words repeatedly at some point, never recovering.
>
> Since this exceeds the 9-page scope of this NeurIPS paper we started working on a forthcoming paper that investigates these questions systematically as it ties strongly into the increasingly important research question of reasoning efficiency (at different reasoning token budgets) which is directly linked to the germane load aspect of schema formulation and tracking in Cognitive Load Theory, and because the generated metadata allows for a very precise analysis of reasoning logic. This is certainly a very clear natural extension of our study that highlights the powerful analysis CogniLoad enables.
>
> To shed more light on this question please see below the proportions of correct answers, failure due to a logic error (within the context limit) and failure because the LLMs reach the maximum context size (32'768 tokens). The figures in the same line as the model name represents the overall proportion across all task lengths per model, while the lines beneath indicate the proportions for the model at the task lengths 20, 50, 100, and 250.
>
> |models & N              |max context reached|logic error|correct|
> |---------------------|-----------|-----------|-------|
> |_o4-mini-2025-04-16_   |_2%_         |_23%_        |_76%_    |
> |20                   |0%         |2%         |98%    |
> |50                   |0%         |8%         |92%    |
> |100                  |0%         |25%        |75%    |
> |250                  |6%         |56%        |38%    |
> |_gemini-2.5-pro_       |_23%_        |_2%_         |_75%_    |
> |20                   |0%         |1%         |99%    |
> |50                   |0%         |1%         |98%    |
> |100                  |14%        |4%         |82%    |
> |250                  |77%        |3%         |21%    |
> |_gemini-2.5-flash_     |_28%_        |_15%_        |_57%_    |
> |20                   |0%         |16%        |84%    |
> |50                   |3%         |18%        |79%    |
> |100                  |28%        |19%        |53%    |
> |250                  |80%        |8%         |12%    |
> |_gemini-2.5-flash-lite_|_1%_         |_60%_        |_39%_    |
> |20                   |0%         |44%        |56%    |
> |50                   |0%         |65%        |35%    |
> |100                  |0%         |66%        |34%    |
> |250                  |5%         |65%        |30%    |
> |_DeepSeek-R1-0528_|_4%_         |_28%_        |_68%_    |
> |20                   |2%         |1%         |97%    |
> |50                   |4%         |9%         |87%    |
> |100                  |6%         |35%        |58%    |
> |250                  |4%         |67%        |29%    |
>
> It is noteworthy how the overall error proportions for the top OpenAI and Google models are mirrored while they have roughly the same proportion of correctly solved problems. Gemini-2.5-pro fails in 23% of all puzzles because it runs out of the context limit while 2% of puzzles fail due to a logic error. Inversely o4-mini has an overall logic error rate of 23% while running out of the context limit in just 2% of the puzzles. We include a discussion of this aspect of model failure in the camera ready version of the paper.
>
> ### *Additional Feedback 3: I wonder how the attribute ontology is selected. For example, will expanding, changing or limiting the category name lead to different results? This contributed to the robustness of the benchmark*
>
> We selected the attribute ontology by choosing natural attributes that pertain to a person (e.g. their location, what they last ate, what they wear on a bodypart) and which can change independently of each other. When we experimented with slightly different ontologies we did not see any obvious change and therefore did not conduct a systematic evaluation.
>
> On expanding or limiting the categories, this is what we do in the “intrinsic difficulty” dimension of CogniLoad:
> This parameter influences the cardinality of the categories in a puzzle and the number of categories in a puzzle (e.g. difficulty 3 means there are three categories that each have a value domain of 3 different values, while a difficulty 10 puzzle uses ten categories with a value domain of 10 values). The number of available categories and their cardinality in the ontology can be changed easily and directly limits the maximum difficulty of puzzles that can be generated for any given ontology.

---

> > ### Comment · Reviewer_CwZ5 · 2025-08-04
> >
> > Thank you for the response. My concerns have been largely addressed. As my expertise primarily lies in mathematical reasoning, I will maintain a confidence score of 2. However, I believe this is a valuable benchmark for evaluating the reasoning capabilities of LLMs, and I will retain my overall score of 5.

---

> > > ### Author Response · Authors · 2025-08-04
> > >
> > > We thank you for your efforts and thoughtful feedback reviewer CwZ5. We are glad that your concerns have been largely addressed and that you perceive our work as a valuable benchmark.

---

> ### Comment · Area_Chair_Sy7t · 2025-08-04
> **Please engage with authors during the discussion period**
>
> Dear reviewer CwZ5,
>
> The authors have responded to your review with a rebuttal to address the three limitations that you noted and respond to your other points of feedback. Please review the rebuttal that they have posted and respond to indicate whether your concerns have been addressed, or, if not, what information the authors should provide or what changes they should make to address them, to the degree that that is possible. There are only a couple of days left in the discussion period, and your engagement during this time is critical.

---

### Decision · Program_Chairs · 2025-09-18

**Decision:**

Reject

**Comment:**

**Summary**: This paper introduces CogniLoad, a synthetic benchmark grounded in Cognitive Load Theory (CLT) that evaluates LLM reasoning using logic puzzles that allow for systematically varying three parameters: task length, intrinsic difficulty, and distractor density, each of which corresponds to different dimensions of CLT. The authors evaluate 14 LLMs and find that task length is the dominant constraint, while also uncovering a distinct U-shaped performance curve in response to distractor information.

**Strengths**:
- Paper is well-written (tRiH, UGjY)
- Using CLT is novel and well-motivated (CwZ5)
- The ability to tune parameters like length and difficulty enables precise & controllable studies (CwZ5, tRiH) with interpretable results (fFRQ)
- Results are relevant for the reasoning capabilities of models (UGjY) and experiments are comprehensive (fFRQ)

**Weaknesses**:
- Limited generalizability as the benchmark only uses a single kind of puzzle (CwZ5). The authors engage with this point during the rebuttal to highlight the benefits of using a “pure deductive reasoning” task, but this is still a limit on the generalizability of the benchmark. Relatedly, fFRQ brings up the lack of linguistic diversity in the benchmark and UGjY brings up the limited definition of a puzzle for this benchmark, both of which could also limit generalizability.
- No testing of proprietary models, even though these represent much of the frontier of sota systems (CZW5, fFRQ, UGjY). The authors use the rebuttal to analyze responses from proprietary models and report the results, and all reviewers reported that their concerns were addressed by these additional results.
- Lack of novelty (tRiH). Authors clarify in the rebuttal that the independence of the tunable parameters in their benchmark is a marked improvement on previous studies, and that their focus on deductive reasoning separates it from other benchmarks.
- Lack of error analysis (CwZ5, tRiH). The authors provide some details during the rebuttal about the errors observed, but state that the findings belong in a separate paper. However, I feel that understanding the pattern of errors that a benchmark reveals is crucial to understanding the benchmark’s utility. At least some discussion of error patterns belongs in the current study.

**Overview / explanation of recommendation**: The benchmark's grounding in Cognitive Load Theory and its ability to independently vary three key parameters make it relevant to assessing the reasoning abilities of LLMs beyond simple performance leaderboards. The primary weakness identified by multiple reviewers was the lack of proprietary, state-of-the-art models in the evaluation. However, the authors provided this in their rebuttal, which included a new set of experiments on five frontier models that demonstrated the benchmark's relevance for sota models and addressed reviewer concerns. While the focus on a single puzzle type limits generalizability, this is a reasonable trade-off for the controllability of the benchmark, though a detailed error analysis would have done an effective job of showcasing why that trade-off was necessary and would have greatly strengthened the paper. However, the current benchmark is a substantial contribution that reviewers agree is useful and meaningful.

===== FINAL UPDATE FROM DB Track PCs ====

The final decision for this paper has been taken by the program chairs after consultation with the SACs. All Senior Area Chairs have ranked papers according to the feedback from the AC during the review process. We decided to leave the original meta-review to reflect the opinion of the AC in light of the initial discussions with reviewers and SAC.